# Chromosome-wide co-fluctuation of stochastic gene expression in mammalian cells

**Mengyi Sun, Jianzhi Zhang** *

Department of Ecology and Evolutionary Biology, University of Michigan, Ann Arbor, MI, United States of America

* jianzhi@umich.edu

**Data Availability Statement:** Software used and the underlying numerical data for all figures can be downloaded from Github (https://github.com/mengysun/Linked_noise).

**Funding:** This work was supported by U.S. National Institutes of Health research grant

## Abstract

Gene expression is subject to stochastic noise, but to what extent and by which means such stochastic variations are coordinated among different genes are unclear. We hypothesize that neighboring genes on the same chromosome co-fluctuate in expression because of their common chromatin dynamics, and verify it at the genomic scale using allele-specific single-cell RNA-sequencing data of mouse cells. Unexpectedly, the co-fluctuation extends to genes that are over 60 million bases apart. We provide evidence that this long-range effect arises in part from chromatin co-accessibilities of linked loci attributable to three-dimensional proximity, which is much closer intra-chromosomally than inter-chromosomally. We further show that genes encoding components of the same protein complex tend to be chromosomally linked, likely resulting from natural selection for intracellular among-component dosage balance. These findings have implications for both the evolution of genome organization and optimal design of synthetic genomes in the face of gene expression noise.

## Author summary

Gene expression is subject to substantial stochastic noise or fluctuation. We hypothesize that expressions of neighboring genes on the same chromosome co-fluctuate because of their common chromatin dynamics. To test this hypothesis, we make use of the fact that each diploid cell contains a maternal and a paternal copy of the same chromosome that are differentiable by their DNA sequences. Hence, allele-specific single-cell RNA-sequencing can quantify the expression level of each allele in individual diploid cells, allowing measuring the expression co-fluctuation of linked alleles as well as that of unlinked alleles of the same genes. Using such data from mouse cells, we discover chromosome-wide gene expression co-fluctuation and provide evidence that this long-range effect arises in part from chromatin co-accessibilities of linked loci attributable to three-dimensional proximity. We show that genes encoding protein complex subunits tend to be chromosomally linked, likely resulting from natural selection for intracellular among-component dosage balance. Thus, minimization of the deleterious effect of gene expression noise has probably produced a nonrandom distribution of genes in the genome. These findings have

GM120093 to J.Z. The funder had no role in study design, data collection and analysis, decision to publish, or preparation of the manuscript.

**Competing interests:** The authors have declared that no competing interests exist.

implications for the evolution of genome organization and optimal design of synthetic genomes.

## Introduction

Gene expression is subject to considerable stochasticity that is known as expression noise, formally defined as the expression variation of a given gene among isogenic cells in the same environment [1–3]. Gene expression noise is a double-edged sword. On the one hand, it can be deleterious because it leads to imprecise controls of cellular behavior, including, for example, destroying the stoichiometric relationship among functionally related proteins and disrupting homeostasis [4–8]. On the other hand, gene expression noise can be beneficial. For instance, unicellular organisms may exploit gene expression noise to employ bet-hedging strategies in fluctuating environments [9, 10], whereas multicellular organisms can make use of expression noise to initiate developmental processes [11–13].

By quantifying protein concentrations in individual isogenic cells cultured in a common environment, researchers have measured the expression noise for thousands of genes in the bacterium *Escherichia coli* [14] and unicellular eukaryote *Saccharomyces cerevisiae* [15]. Nevertheless, because genes are not in isolation, one wonders whether and to what extent expression levels co-vary among genes at a steady state, which unfortunately cannot be studied by the above data. By simultaneously tagging two genes with different florescent markers, Stewart-Ornstein et al. discovered strong co-fluctuation of the concentrations of some functionally related proteins in yeast such as those involved in the Msn2/4 stress response pathway, amino acid synthesis, and mitochondrial maintenance, respectively [16], and the expression co-fluctuation of these genes is facilitated by their sharing of transcriptional regulators [17].

Here we explore yet another mechanism for expression co-fluctuation. We hypothesize that, due to the sharing of chromatin dynamics [18], a key contributor to gene expression noise [18–20], genes that are closely linked on the same chromosome should exhibit a stronger expression co-fluctuation when compared with genes that are not closely linked or unlinked (Fig 1). We refer to this potential influence of chromosomal linkage of two genes on their expression co-fluctuation as the linkage effect. The linkage-effect hypothesis is supported by two pioneering studies demonstrating that the correlation in expression level between two reporter genes across isogeneic cells in the same environment is much higher when they are placed next to each other on the same chromosome than when they are placed on separate chromosomes [21, 22]. However, neither the generality of the linkage effect nor the chromosomal proximity required for this effect are known. Furthermore, the biological significance of the linkage effect and its potential impact on genome organization and evolution have not been investigated. In this study, we address these questions by analyzing allele-specific single-cell RNA-sequencing (RNA-seq) data from mouse cells [23]. We demonstrate that the linkage effect is not only general but also long-range, extending to genes that are tens of millions of bases apart. We provide evidence that three-dimensional (3D) chromatin proximities are responsible for the long-range expression co-fluctuation through mediating chromatin accessibility covariations. Finally, we show theoretically and empirically that the linkage effect has likely impacted the evolution of the chromosomal locations of genes encoding members of the same protein complex.

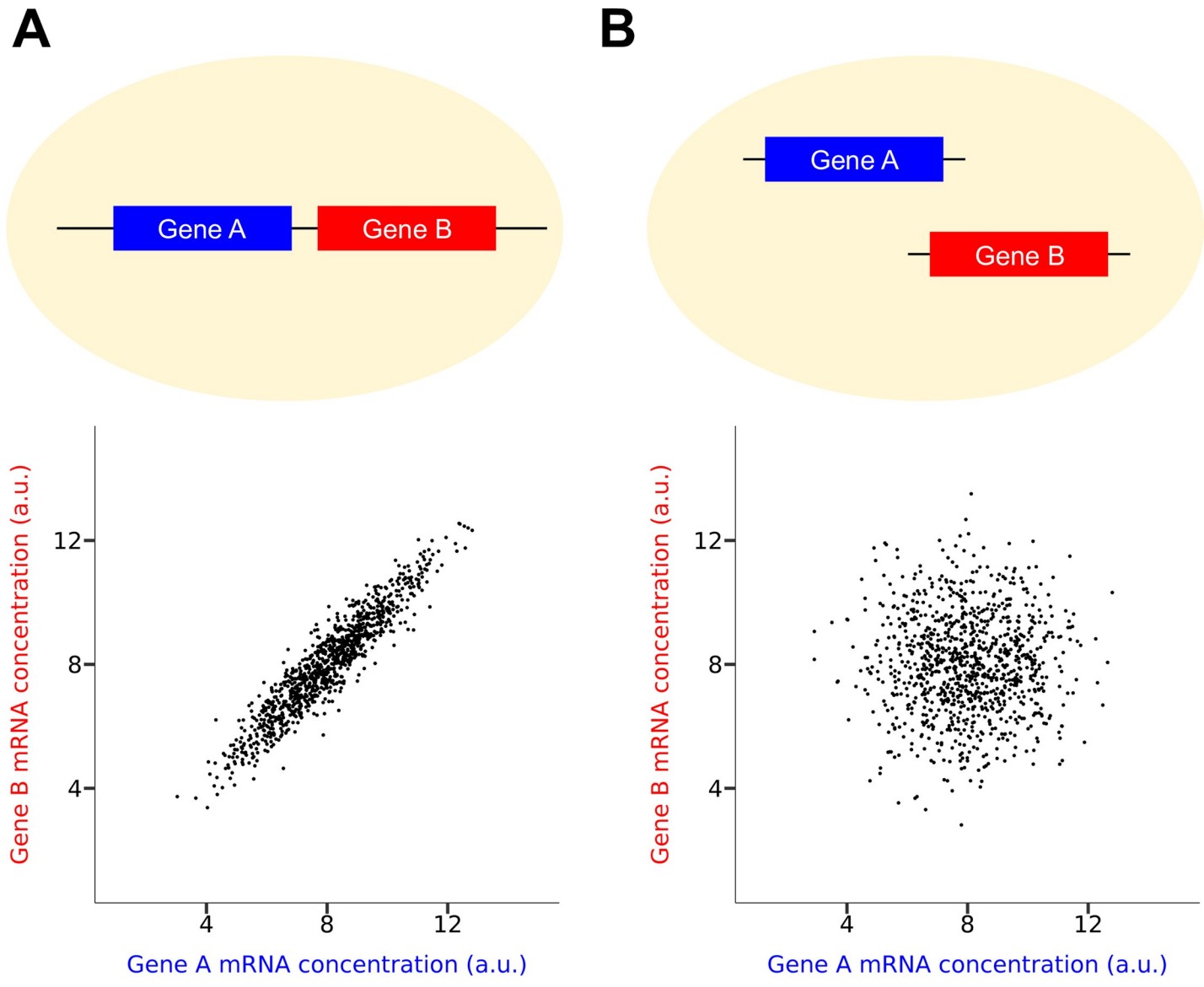

**Fig 1. The hypothesized linkage effect on gene expression co-fluctuation.** The cellular mRNA concentrations of two genes should be better correlated among isogenic cells in a population under a constant environment (A) when the two genes are chromosomally linked than (B) when they are unlinked. In the dot plot, each dot represents a cell.

## Results

### Linkage effect on gene expression co-fluctuation is general and long-range

Let us consider two genes $A$ and $B$ each with two alleles respectively named 1 and 2 in a diploid cell. When $A$ and $B$ are chromosomally linked, without loss of generality, we assume that $A_1$ and $B_1$ are on the same chromosome whereas $A_2$ and $B_2$ are on its homologous chromosome (Fig 2A). Expression co-fluctuation between one allele of $A$ and one allele of $B$ (e.g., $A_1$ and $B_2$) is measured by Pearson's correlation ($r_e$, where the subscript "e" stands for expression) between the expression levels of the two alleles across isogenic cells under the same environment. Among the four possible pairs of an $A$ and a $B$ allele, $A_1$-$B_1$, $A_2$-$B_2$, $A_1$-$B_2$, and $A_2$-$B_1$, the former two pairs are physically linked whereas the latter two pairs are unlinked. The linkage-

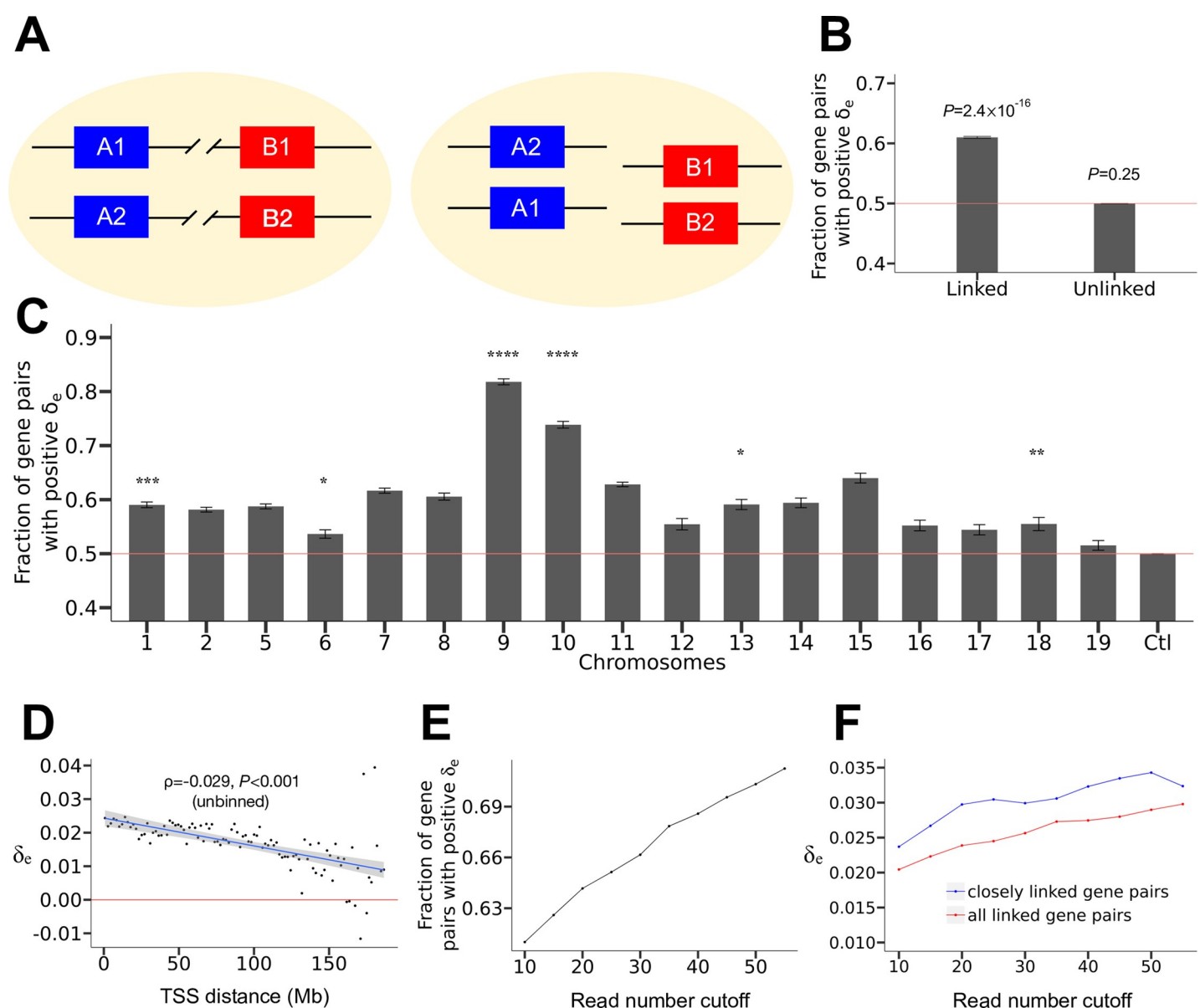

**Fig 2. Chromosome-wide linkage effects on gene expression co-fluctuation in mouse fibroblast cells.** (A) The logic of the method for testing the linkage effect. When gene *A* and gene *B* are linked, the correlations between the mRNA concentrations of the alleles of *A* and *B* that are physically linked (*cis*-correlations) should exceed the corresponding correlations of the alleles that are physically unlinked (*trans*-correlations). That is, $\delta_e$ = (sum of *cis*-correlations − sum of *trans*-correlations)/2 should be positive. This relationship should disappear if gene *A* and gene *B* are unlinked. For each gene, the paternal and maternal alleles are labeled by 1 and 2, respectively. (B) Fraction of gene pairs with positive $\delta_e$. The red line represents the null expectation under no linkage effect. The 95% confidence intervals estimated from all gene pairs are presented. *P*-values from binomial tests on independent gene pairs are presented. (C) Fraction of gene pairs with positive $\delta_e$ in each chromosome. The 95% confidence intervals estimated from all gene pairs are presented. Binomial *P*-values estimated from independent gene pairs are indicated as follows. NS, not significant; *, $0.01 < P < 0.05$; **, $0.001 < P < 0.01$; ***, $0.0001 < P < 0.001$; and ****, $P < 0.0001$. The red line represents the null expectation under no linkage effect. The control (Ctl) shows the fraction of unlinked gene pairs with positive $\delta_e$. (D) Median $\delta_e$ in a bin decreases with the median genomic distance of linked genes in the bin. All bins have the same genomic distance interval. TSS, transcription start site. The red line shows median $\delta_e = 0$. The blue line shows the linear regression of the binned data, and the 95% confidence interval of the regression is presented. Spearman's ρ from unbinned data and associated *P*-value determined by a shuffling test are presented. (E) Fraction of linked gene pairs showing positive $\delta_e$ increases with the minimal number of reads per allele required ($P < 10^{-300}$). (F) Median $\delta_e$ for all linked gene pairs (red) and median $\delta_e$ in the left-most bin of panel D (blue) increase with the minimal read number per allele required (both $P < 10^{-300}$).

effect hypothesis asserts that, at a steady state, expression correlations between linked alleles (*cis*-correlations) are greater than those between unlinked alleles (*trans*-correlations). That is,

$\delta_e = [r_e(A_1,B_1)+r_e(A_2,B_2)-r_e(A_1,B_2)-r_e(A_2,B_1)]/2>0$. Note that this formulation is valid regardless of whether the two alleles of the same gene have equal mean expression levels. While each of the four correlations could be positive or negative, in the large data analyzed below, they are mostly positive and show approximately normal distributions across gene pairs examined (S1 Fig).

To verify the above prediction about $\delta_e$, we analyzed a single-cell RNA-seq dataset of fibroblast cells derived from a hybrid between two mouse strains (CAST/EiJ × C57BL/6J) [23]. Single-cell RNA-seq profiles the transcriptomes of individual cells, allowing quantifying stochastic gene expression variations among isogenic cells in the same environment [24–26]. DNA polymorphisms in the hybrid allow estimation of the expression level of each allele for thousands of genes per cell. The dataset includes data from seven fibroblast clones and some non-clonal fibroblast cells of the same genotype. We focused our analysis on clone 7 (derived from the hybrid of CAST/EiJ male × C57BL/6J female) in the dataset, because the number of cells sequenced in this clone is the largest ($n = 60$) among all clones. We excluded from our analysis all genes on Chromosomes 3 and 4 due to aneuploidy in this clone and X-linked genes due to X inactivation. To increase the sensitivity of our analysis and remove imprinted genes, we focused on the 3405 genes that have at least 10 RNA-seq reads averaged across cells mapped to each of the two alleles. Note that this gene set constitute >30% of all expressed genes in the cells concerned. While most of the 3405 genes tend to be highly expressed, 16% of them have lower expressions than the median expression of all expressed genes. The 3405 genes form $3404 \times 3405/2 = 5,795,310$ gene pairs, among which 377,584 pairs are chromosomally linked.

For each pair of chromosomally linked genes, we computed their $\delta_e$ by treating the allele from CAST/EiJ as allele 1 and that from C57BL/6J as allele 2 at each locus. The fraction of gene pairs with $\delta_e > 0$ is 0.61 (Fig 2B). This fraction has a rather narrow 95% confidence interval (Fig 2B), demonstrating that the fraction is significantly higher than the null expectation of 0.5. Because a gene can appear in multiple gene pairs, which are not mutually independent, we applied a binomial test in a subset of gene pairs where each gene appears only once. Specifically, we randomly shuffled the relative positions of all genes on each chromosome and considered from one end of the chromosome to the other end non-overlapping consecutive windows of two genes. The observed fraction of gene pairs with positive $\delta_e$ still significantly exceeds the null expectation of 0.5 ($P < 2.4 \times 10^{-16}$, binomial test). That most gene pairs exhibit $\delta_e > 0$ holds in each of the 17 chromosomes examined, with the trend being statistically significant in six chromosomes even by the above conservative test (nominal $P < 0.05$; Fig 2C). As a negative control, we analyzed gene pairs located on different chromosomes, treating alleles the same way as described above. As expected, this time the fraction of gene pairs with $\delta_e > 0$ is not significantly different from 0.5 ($P = 0.25$; Fig 2B). The fraction of gene pairs with $\delta_e > 0$ appears to vary among chromosomes (Fig 2C). To assess the significance of this variation, we compared the fraction of independent gene pairs with $\delta_e > 0$ between every two chromosomes by Fisher's exact test. After correcting for multiple testing, we found no significant difference between any two chromosomes.

To examine the generality of the findings from clone 7, we also analyzed clone 6 (derived from the hybrid of CAST/EiJ male × C57BL/6J female), which has 38 cells with RNA-seq data. Because 10 of these cells are aneuploidy for different chromosomes (see the supplementary materials in [23]), we analyzed the remaining 28 cells. Similar results were obtained (S2A and S2B Fig). Because clone 6 was from a male whereas clone 7 was from a female, our results apparently apply to both sexes. We also analyzed 47 non-clonal fibroblast cells with the same genetic background (cell IDs from 124 to 170, derived from the hybrid of CAST/EiJ male × C57BL/6J female), and obtained similar results (S2C and S2D Fig). These findings establish that the linkage effect on expression co-fluctuation is neither limited to a few genes in

a specific clone nor an epigenetic artifact of clonal cells, but is general. The linkage effect on co-fluctuation (and the decrease of the effect with genomic distance shown below) is robust to the definition of $\delta_e$, because similar results are obtained when correlation coefficients are replaced with squares of correlation coefficients in the definition of $\delta_e$.

We next investigated how close two genes need to be on the same chromosome for them to co-fluctuate in expression. We divided all pairs of chromosomally linked genes into 100 equal-interval bins based on the genomic distance between genes, defined by the number of nucleotides between their transcription start sites (TSSs). The median $\delta_e$ in a bin is found to decrease with the genomic distance represented by the bin (Fig 2D). Furthermore, even for the unbinned data, $\delta_e$ for a pair of linked genes correlates negatively with their genomic distance (Spearman's $\rho$ = -0.029). To assess the statistical significance of this negative correlation, we randomly shuffled the genomic coordinates of genes within chromosomes and recomputed the correlation. This was repeated 1000 times and none of the 1000 $\rho$ values were equal to or more negative than the observed $\rho$. Hence, the linkage effect on expression co-fluctuation of two linked genes weakens significantly with their genomic distance ($P < 0.001$).

Surprisingly, however, median $\delta_e$ exceeds 0 for every bin except when the genomic distance exceeds 150 Mb (Fig 2D). Hence, the linkage effect is long-range. To statistically verify the potentially chromosome-wide linkage effect, we focused on linked gene pairs that are at least 63 Mb apart, which is one half the median size of mouse chromosomes. The median $\delta_e$ for these gene pairs is 0.017, or 68% of the median $\delta_e$ for the left-most bin in Fig 2D. We randomly shuffled the genomic positions of all genes and repeated the above analysis 1000 times. In none of the 1000 shuffled genomes did we observe the median $\delta_e$ greater than 0.017 for linked genes of distances >63 Mb, validating the long-range expression co-fluctuation in the actual genome. The same can be said even for linked genes of distances >90 Mb ($P < 0.001$, shuffling test). The above observations are not clone-specific, because the same trend is observed for cells of clone 6 (S2B Fig).

Notably, a previous experiment in mammalian cells [21] detected a linkage effect for chromosomally adjacent reporter genes ($\delta_e$ = 0.834) orders of magnitude stronger than what is observed here. This is primarily because expression levels estimated using single-cell RNA fluorescence in situ hybridization in the early study [21] are much more precise than those estimated using allele-specific single-cell RNA-seq [27] here. We thus predict that the linkage effect detected will be more pronounced as the expression level estimates become more precise. As a proof of principle, we gradually raised the required minimal number of reads per allele in our analysis, which should increase the precision of expression level estimation but decrease the number of genes that can be analyzed. Indeed, as the minimal read number rises, the fraction of chromosomally linked gene pairs with a positive $\delta_e$ (Fig 2E), median $\delta_e$ for all chromosomally linked gene pairs (Fig 2F), and median $\delta_e$ for the left-most bin (Fig 2F) all increase significantly. Furthermore, the low capturing efficiency of single-cell RNA-seq substantially reduces the observed size of the linkage effect, and our lower-bound estimate of the median true $\delta_e$ is 0.15 (see Materials and Methods).

Because what matters to a cell is the total number of transcripts produced from the two alleles of a gene instead of the number produced from each allele, we also calculated the pairwise correlation in expression level between genes using either the total number of reads mapped to both alleles of a gene or normalized expression level of the gene. We similarly found a long-range linkage effect (S3 Fig), with trends and effect sizes close to the observations based on allele-specific expressions.

Previous studies reported that the relative transcriptional orientations of neighboring genes influence their expression co-fluctuation [28]. This impact, however, is unobserved in our

study (S4 Fig), which may be due to the limited precision of the expression estimates and the fact that only 422 pairs of neighboring genes satisfy the minimal read number requirement.

## Shared chemical environment for transcription results in the long-range linkage effect

What has caused the chromosome-wide expression co-fluctuation of linked genes? One simple explanation is the asynchronous DNA replication in dividing cells, where closely linked genes tend to be replicated at the same time so show positively correlated gene copy numbers and hence expression levels. But a simple calculation demonstrates that this explanation is not tenable. There are $10^4$ to $10^5$ replication origins per mammalian cell [29]. Given the size of the mammalian genome (~$3\times10^9$ bases), DNA segments within 0.03–0.3 Mb share a replication origin. The asynchronous DNA replication could result in the expression co-fluctuation of genes in the range of 0.03 to 0.3 Mb, which cannot explain our observation of expression co-fluctuation at the scale of >60 Mb.

Individual chromosomes in mammalian cells are organized into territories with a diameter of 1~2 μm [30], whereas the diameter of the nucleus is ~8 μm [30]. Thus, the physical distance between chromosomally linked genes is below 1~2 μm, whereas that between unlinked genes is usually > 1~2 μm and can be as large as ~8 μm. Because it takes time for macromolecules to diffuse in the nucleus, linked genes tend to have similar chemical environments and hence similar transcriptional dynamics (i.e., promoter co-accessibility and/or co-transcription) when compared with unlinked genes. We thus hypothesize that the linkage effect is fundamentally explained by the 3D proximity of linked genes compared with unlinked genes (Fig 3A). Below we provide evidence for this model. Note, while our computational analyses cannot prove causation, they examine correlations among various quantities that are predicted by our hypothesis and hence can provide strong evidence for or against the hypothesis when performed rigorously and interpreted appropriately.

We started by comparing the 3D distances between linked alleles with those between unlinked alleles. The 3D distance between two genomic regions can be approximately measured by Hi-C, a high-throughput chromosome conformation capture method for quantifying the number of interactions between genomic loci that are nearby in 3D space [31]. The smaller the 3D distance between two genomic regions, the higher the interaction frequency between them [32]. It is predicted that the interaction frequency between the physically linked alleles of two genes (*cis*-interaction) is greater than that between the unlinked alleles of the same gene pair (*trans*-interaction). To verify this prediction, we analyzed the recently published allele-specific 500kb-resolution Hi-C interaction matrix [33] of mouse neural progenitor cells (NPC). For any two linked loci $A$ and $B$ as depicted in the left diagram of Fig 2A, we computed $\delta_i = [F(A_1,B_1)+F(A_2,B_2)-F(A_1,B_2)-F(A_2,B_1)]/2$, where $F$ is the interaction frequency between the two alleles in the parentheses and the subscript "i" refers to interaction. We found that 99% of pairs of linked loci have a positive $\delta_i$ ($P < 2.2\times10^{-16}$, binomial test on independent locus pairs; Fig 3B). By contrast, among unlinked gene pairs, the fraction with a positive $\delta_i$ is not significantly different from that with a negative $\delta_i$ ($P = 0.90$, binomial test on independent locus pairs; Fig 3B). In the analysis of unlinked loci, we treated all alleles from one parental strain of the hybrid as alleles 1 and all alleles from the other parental strain of the hybrid as alleles 2 in the above formula of $\delta_i$. These results clearly demonstrate the 3D proximity of genes on the same chromosome when compared with those on two homologous chromosomes.

To examine if the above phenomenon is long-range, we plotted $\delta_i$ as a function of the distance (in Mb) between two linked loci considered. Indeed, even when the distance exceeds 63 Mb, one half the median size of mouse chromosomes, almost all locus pairs still show positive

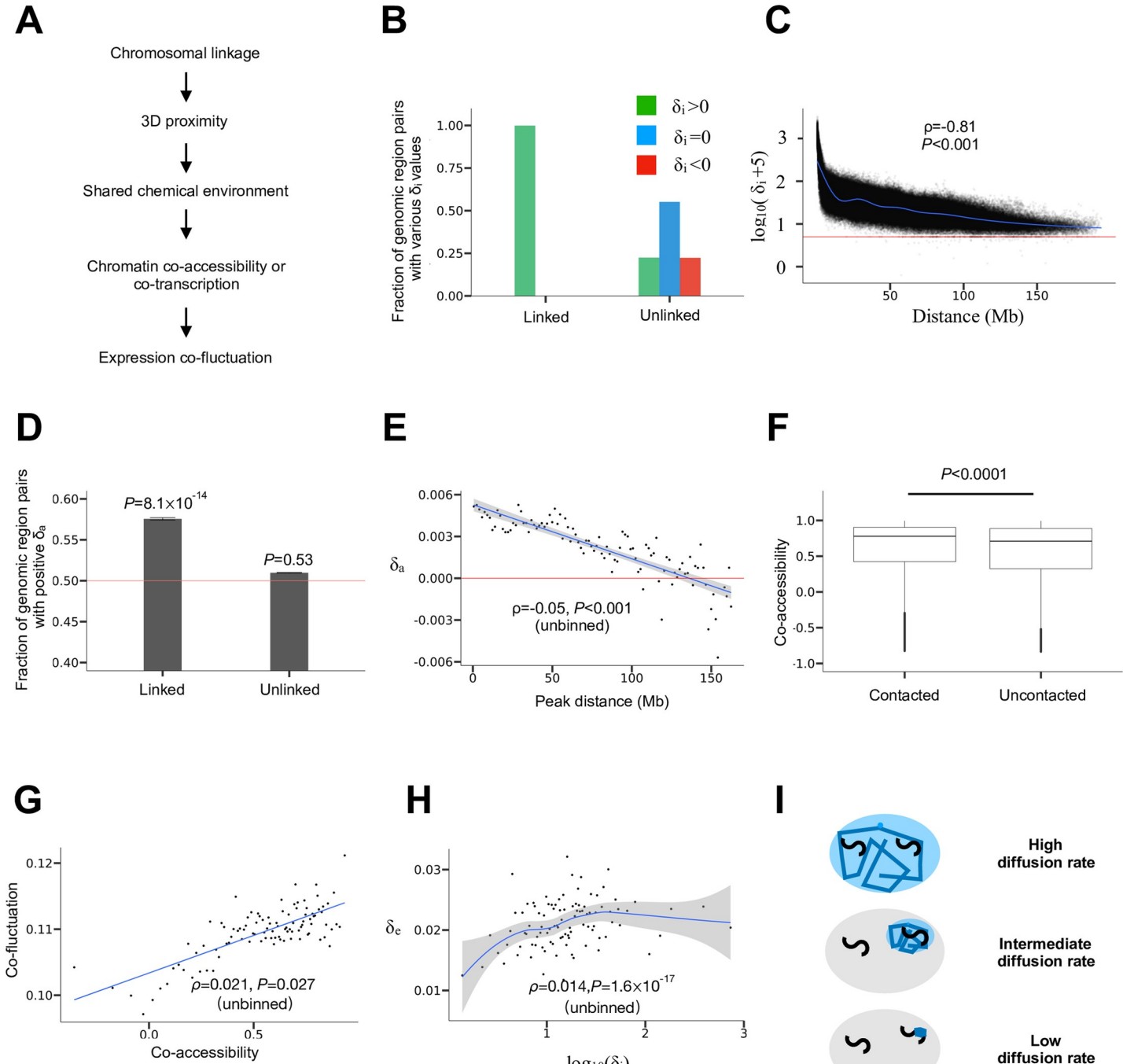

**Fig 3. Mechanistic basis of the linkage effect on expression co-fluctuation.** (A) A model on how chromosomal linkage causes expression co-fluctuation. (B) Fractions of linked or unlinked genomic region pairs with positive, 0, and negative $\delta_i$ values, respectively. $\delta_i$ = (sum of *cis*-interactions − sum of *trans*-interactions)/2, where chromatin interactions are based on Hi-C data. All fractions are shown, but the blue and red bars for linked regions are too low to be visible. (C) $\delta_i$ decreases with the genomic distance between the linked regions considered. Each dot represents one pair of linked genomic regions. Shown here is $\log_{10}(\delta_i + 5)$ because $\delta_i$ is occasionally negative and it decreases with genomic distance very quickly. The horizontal red line indicates $\delta_i = 0$. The blue line is a cubic spline regression of $\log_{10}(\delta_i + 5)$ on the genomic distance. Spearman's $\rho$ from unbinned data and associated *P*-value determined by a shuffling test are presented. (D) Fraction of linked or unlinked pairs of ATAC peaks with positive $\delta_a$. $\delta_a$ = (sum of *cis*-correlations in accessibility − sum of *trans*-correlations in accessibility)/2. The 95% confidence interval is presented. *P*-values from binomial tests on independent peak pairs are presented. The red line shows the fraction of 0.5. (E) $\delta_a$ decreases with the distance between linked ATAC peaks. Each dot represents a bin. All bins have the same distance interval. The red line shows $\delta_a = 0$. The blue line shows the linear regression of the binned data and the shade shows the 95% confidence interval of the regression. For better viewing, one bin (X = 156, Y = -0.02) is omitted in the figure; the extreme $\delta_a$ of the bin is probably due to the small sample size of the bin (*n* = 13). Spearman's $\rho$ computed from unbinned data and associated *P*-value determined from a shuffling test are presented. (F)

Co-accessibility (*trans-r*$_a$) is greater for 3D contacted (*trans-F* > 0) than uncontacted (*trans-F* = 0) non-allelic genomic regions located on homologous chromosomes. The lower and upper edges of a box represent the first (qu$_1$) and third (qu$_3$) quartiles, respectively, the horizontal line inside the box indicates the median (md), the whiskers extend to the most extreme values inside inner fences, md±1.5(qu$_3$-qu$_1$), and the dots represent values outside the inner fences (outliers). *P*-value is determined by a Mantel test. (G) Expression co-fluctuation (*trans-r*$_e$) improves with the co-accessibility (*trans-r*$_a$) of non-allelic ATAC peaks located on homologous chromosomes. Each dot represents a bin. All bins have the same co-accessibility interval. The blue line shows the linear regression of the binned data. Spearman's ρ computed from unbinned data and associated *P*-value determined by a Mantel test are presented. (H) $\delta_e$ is positively correlated with $\delta_i$. All gene pairs are separated into 100 equal-size bins, and the blue line shows a LOESS regression with the 95% confidence interval of the regression shown. Spearman's ρ computed from unbinned data and the associated *P*-value are presented. (I) Diffusion rates for molecules responsible for the chromosome-wide linkage effect should be neither too high nor too low. If the diffusion is too fast, the concentration of the molecule will be similar across the nucleus (top); if the diffusion is too slow, the concentration cannot even be similar for loci loosely linked on the same chromosome (bottom). Only when the diffusion rate is intermediate could the local chemical environment be homogeneous for genes on the same chromosome but heterogeneous for genes on different chromosomes (middle). The large oval represents the nucleus and each black "S" curve represents a chromosome. Blue zig-zags show molecular diffusions, while the blue area depicts a chemically homogenous environment.

$\delta_i$ (Fig 3C). Similar to the phenomenon of the linkage effect on gene expression co-fluctuation, we observed a negative correlation between the genomic distance between two linked loci and $\delta_i$ (ρ = -0.81 for unbinned data). This correlation is statistically significant (*P* < 0.001), because it is stronger than the corresponding correlation in each of the 1000 negative controls where the genomic positions of all genes are randomly shuffled within chromosomes.

As mentioned, 3D proximity should synchronize the transcriptional dynamics of linked alleles. Based on the bursty model of gene expression [34], transcription involves two primary steps. In the first step, the promoter region switches from the inactive state to the active state such that it becomes accessible to the transcriptional machinery. In the second step, RNA polymerase binds to the activated promoter to initiate transcription. In principle, the synchronization of either step can result in co-fluctuation of mRNA concentrations. Because the accessibility of promoters can be detected using transposase-accessible chromatin using sequencing (ATAC-seq) [35] in a high-throughput manner, we focused our empirical analysis on promoter co-accessibility. Note that although the bursty model does not consider certain details of transcription such as polymerase pausing and productive elongation, there is mounting evidence that the bursty model provides a good approximation of stochastic gene expression [21, 36–38], which is what matters to our study.

To verify the potential long-range linkage effect on chromatin co-accessibility, we should ideally use single-cell allele-specific measures of chromatin accessibility. However, such data are unavailable. We reason that, the accessibility covariation of genomic regions among cells may be quantified by the corresponding covariation among populations of cells of the same type cultured under the same environment. In fact, it can be shown mathematically that, under certain conditions, chromatin co-accessibility of two genomic regions among cells equals the corresponding chromatin co-accessibility across cell populations (see Materials and Methods). Based on this result, we analyzed a dataset collected from allele-specific ATAC-seq in 16 NPC cell populations [39]. We first removed sex chromosomes and then required the number of reads mapped to each allele of a peak to exceed 50 for the peak to be considered. This latter step removed imprinted loci and ensured that the considered peaks are relatively reliable. About 3500 peaks remained after the filtering. This sample size is comparable to the number of genes used in the analysis of expression co-fluctuation. For each pair of ATAC peaks, we computed $\delta_a = [r_a(A_1,B_1)+r_a(A_2,B_2)−r_a(A_1,B_2)−r_a(A_2,B_1)]/2$, where $r_a$ is the correlation in ATAC-seq read number between the alleles specified in the parentheses (following the left diagram in Fig 2A) across the 16 cell populations and the subscript "a" refers to chromatin accessibility. The fraction of peak pairs with a positive $\delta_a$ is significantly greater than 0.5 for linked peak pairs but not significantly different from 0.5 for unlinked peak pairs (binomial test on independent peak pairs; Fig 3D). Furthermore, after grouping ATAC peak pairs into 100 equal-interval bins according to the genomic distance between peaks, we observed a clear trend that $\delta_a$ decreases with the genomic distance between peaks (*ρ* = -0.05 for unbinned data,

$P < 0.001$, within-chromosome shuffling test; Fig 3E). In addition, even for linked peak pairs with a distance greater than 63 Mb, their median $\delta_a$ is significantly greater than that of unlinked peak pairs ($P < 0.001$, among-chromosome shuffling test). Together, these results demonstrate a long-range linkage effect on chromatin co-accessibility. Similar to $\delta_e$, the observed $\delta_a$ is small. This is again at least in part a result of low capturing efficiencies in high-throughput sequencing. We estimated that the median true $\delta_a$ is at least 0.03 (see Materials and Methods), an order of magnitude larger than the observed value.

Because we hypothesize that the linkage effect on expression co-fluctuation is via 3D chromatin proximity that leads to chromatin co-accessibility (Fig 3A), we should verify the relationship between 3D proximity and chromatin co-accessibility for unlinked genomic regions to avoid the confounding factor of linkage. To this end, we converted ATAC-seq read counts to a 500kb resolution by summing up read counts for all allele-specific chromatin accessibility peaks that fall within the corresponding Hi-C bin, because the resolution of the Hi-C data is 500kb. Because alleles from different parents are unlinked in the hybrid used for ATAC-seq, for each pair of bins, we computed the mean correlation in chromatin accessibility between the alleles derived from different parents among the 16 cell populations, or $trans\text{-}r_a = r_a(A_1, B_2)/2 + r_a(A_2, B_1)/2$. For the same reason, we computed the sum of Hi-C contact frequency between the alleles derived from different parents, $trans\text{-}F = F(A_1, B_2)/2 + F(A_2, B_1)/2$. Because interaction frequencies in Hi-C data are generally low for unlinked regions, we separated all pairs of bins into two categories, contacted (i.e., $trans\text{-}F > 0$) and uncontacted (i.e., $trans\text{-}F = 0$). We found that $trans\text{-}r_a$ values for contacted bin pairs are significantly higher than those for uncontacted bin pairs ($P < 0.0001$; Fig 3F), consistent with our hypothesis that 3D chromatin proximity induces chromatin co-accessibility. The above statistical significance was determined by performing a Mantel test using the original $trans\text{-}r_a$ matrix of the aforementioned allele pairs and the corresponding $trans\text{-}F$ matrix. Corroborating our finding, a recent study of single-cell (but not allele-specific) chromatin accessibility data also found that the co-accessibility of two loci rises with their 3D proximity [40].

To test the hypothesis that chromatin co-accessibility leads to expression co-fluctuation (even for unlinked alleles) (Fig 3A), we analyzed the allele-specific ATAC-seq data and single-cell allele-specific RNA-seq data together. Although these data were generated from different cell types in mouse, we reason that, because the 3D chromosome conformation is highly similar among tissues [41], chromatin co-accessibility, which is affected by 3D chromatin proximity (Fig 3F), may also be similar among tissues. Hence, it may be possible to detect a correlation between chromatin co-accessibility and expression co-fluctuation. To this end, we used unbinned ATAC-peak data to compute $trans\text{-}r_a$ but limited the analysis to those peaks with at least 10 reads per allele. We used the allele-specific RNA-seq data to compute $trans\text{-}r_e = r_e(A_1, B_2)/2 + r_e(A_2, B_1)/2$ for pairs of linked genes. We then assigned each gene to its nearest ATAC peak and averaged $trans\text{-}r_e$ among gene pairs assigned to the same pair of ATAC peaks. We subsequently grouped ATAC peak pairs into 100 equal-interval bins according to their co-accessibilities, and observed a clear positive correlation between median $trans\text{-}r_a$ and median $trans\text{-}r_e$ across the 100 bins (Fig 3G). For unbinned data, $trans\text{-}r_a$ and $trans\text{-}r_e$ also show a significant, positive correlation ($\rho = 0.021$, $P = 0.027$, Mantel test). Although the assignment of a gene to its nearest ATAC peak may not be biologically meaningful in some cases, such potential errors only add noise to our analysis, meaning that the true signal should be stronger than what is observed.

As predicted, there is also a positive correlation between $\delta_i$ and $\delta_e$ ($\rho = 0.014$, $P = 1.6 \times 10^{-17}$). In this analysis, for each linked pair of Hi-C bins, we computed $\delta_i$. We assigned each gene in our dataset to its nearest Hi-C bin, estimated $\delta_e$ for each pair of genes that are respectively located in the two Hi-C bins considered, and computed the average $\delta_e$ between the two bins.

The correlation between $\delta_i$ and $\delta_e$ can be visualized in Fig 3H, where Hi-C bin pairs are separated into 100 equal-size groups based on $\delta_i$. Note that although the impact of 3D proximity (Fig 3H) appears weaker than the impact of genomic distance (Fig 2D) on $\delta_e$, these two plots are not directly comparable because of the following reasons. First, the Hi-C contact frequency is not an accurate measure of 3D proximity, especially for region pairs that rarely contact, which apply to the vast majority of region pairs (Fig 3C). By contrast, genomic distance is measured accurately. Second, the resolution of the Hi-C data used is much lower (500 kb) than that of the genomic distance (1 bp). Third, the Hi-C data and gene expression co-fluctuations data are not from the same cell type, which reduces the observed impact of 3D proximity.

Together, the above results support our hypothesis that, compared with unlinked genes, linked genes have a shared chemical environment due to their 3D proximity and hence chromatin co-accessibility, which leads to their expression co-fluctuation (Fig 3A). However, 3D proximity can lead to promoter co-accessibility by several means, which have been broadly summarized into three categories of mechanisms [30]: 1D scanning, 3D looping, and 3D diffusion. 1D scanning refers to the spread of chromatin states along an entire chromosome, but 1D scanning is rare, with only a few known examples such as X-chromosome inactivation [30]. Hence, 1D scanning is unlikely to be the mechanism responsible for the broad linkage effect discovered here. 3D looping refers to the phenomenon that a chromosome often forms loops to bring far-separated loci into contact, whereas 3D diffusion refers to chromosome communication by local diffusion of transcription-related proteins. For tightly linked loci, our data do not allow a clear distinction between 3D looping and 3D diffusion in causing the linkage effect discovered here. But 3D diffusion seems more likely for the long-range effect, because the range of 3D looping seems limited to loci separated by no more than 200 kb simply due to the rapid decrease of the contact frequency with the physical distance between two loci [42], evident in Fig 3C (note the log scale of the Y-axis). It has been estimated that loci separated by 10 Mb behave essentially the same as two loci that are on different chromosomes in terms of the contact frequency [30], and any contact-based mechanism is unlikely to be long-range (e.g., topologically associating domains) [41]. Therefore, the most likely cause of our observed long-range linkage effect is 3D diffusion.

In the 3D diffusion mechanism, which molecule is most likely responsible for the observed long-range linkage effect on expression co-fluctuation? If the chemical influencing transcription has a diffusion time in the nucleus much shorter than the interval between transcriptional bursts, two genes have essentially the same environment with respect to that chemical regardless of their 3D distance [43] and hence no linkage effect is expected (top cell in Fig 3I). On the contrary, if the chemical diffuses too slowly, the linkage effect will be local [43] and hence cannot be chromosome-wide (bottom cell in Fig 3I). Therefore, the diffusion rate of the chemical responsible for the long-range linkage effect cannot be too low or too high such that they become evenly distributed in a chromosome territory but not the whole nucleus in a time comparable to the interval between transcriptional bursts (middle cell in Fig 3I). The typical transcriptional burst interval is 18–50 minutes in mammalian cells [37, 38]. The time for a chemical to distribute evenly in a given volume with radius $R$ is on the order of $R^2/D$, where $D$ is the diffusion coefficient of the chemical [34]. Most molecules in the nucleus are rapidly diffused. For example, transcription factors typically have a diffusion coefficient of 0.5–5 $\mu m^2$/s in the nucleus [34, 44], meaning that they can diffuse across the whole nucleus in about 3 to 30 seconds. By contrast, core histone proteins such as H2B proteins diffuse extremely slowly due to their tight binding to DNA. They are usually considered immobilized because diffusion is rarely observed during the course of an experiment [44, 45]. Therefore, none of these molecules are responsible for the long-range linkage effect observed. Interestingly, linker histones,

which include five subtypes of H1 histones in mouse that play important roles in chromatin structure and transcription regulation [46], have a diffusion coefficient of about 0.01 $\mu m^2$/s [47]. Thus, it takes H1 proteins 25–100 seconds to diffuse through a chromosome territory, but about 30 minutes to diffuse across the whole nucleus. The former time but not the latter is much smaller than the typical transcriptional burst interval. Hence, it is possible that H1 diffusion in the nucleus is the ultimate cause of the linkage effect. We provide empirical evidence for this hypothesis in a later section.

## Beneficial linkage of genes encoding components of the same protein complex

Our finding that chromosomal linkage leads to gene expression co-fluctuation implies that linkage between genes could be selected for when expression co-fluctuation is advantageous. Due to the complexity of biology, it is generally difficult to predict whether the expression co-fluctuation of a pair of genes is beneficial, neutral, or deleterious. However, the expression co-fluctuation of genes encoding components of the same protein complex is likely advantageous. To see why this is the case, let us consider a dimer composed of one molecule of protein A and one molecule of protein B; the heterodimer is functional but monomers are not. We denote the concentration of dissociated protein A as [A], the concentration of dissociated protein B as [B], and the concentration of protein complex AB as [AB]. At the steady state, [AB] = $K$[A][B], where $K$ is the association constant [48]. Furthermore, the total concentration of protein A, $[A]_t$, equals [A] + [AB], while the total concentration of protein B, $[B]_t$, equals [B] + [AB]. Based on these relationships, we simulated 10,000 cells, where the mean and coefficient of variation (CV) are respectively 1 and 0.2 for both $[A]_t$ and $[B]_t$ (see Materials and Methods). We assumed $K = 10^5$ based on empirical $K$ values of stable protein complexes [49]. We found that, as the correlation between $[A]_t$ and $[B]_t$ increases, mean [AB] of the 10,000 cells rises (Fig 4A). We also considered a wide range of other $K$ values ($10^{-1}$, $10^0$, $10^1$, $10^2$, $10^3$, and $10^4$) and found the result largely unchanged. The lower-bound mean [AB] is about 3% higher under co-fluctuation than under no co-fluctuation. Furthermore, the effect size rises substantially with CV. For example, when CV = 0.5, which is not unusual in eukaryotes [49], the effect increases to 20%. We also considered protein complexes with other stoichiometries and the scenario when the mean $[A]_t$ to mean $[B]_t$ ratio deviates from the stoichiometry (see Materials and Methods). In all parameter combinations examined, the mean [AB] increases with the correlation between $[A]_t$ and $[B]_t$, albeit with a wide range of effect size (0.001% to 27% higher mean [AB] under co-fluctuation than under no co-fluctuation). If we assume that fitness rises with [AB], the co-fluctuation of $[A]_t$ and $[B]_t$ is beneficial, compared with independent fluctuations of $[A]_t$ and $[B]_t$. In addition, because mean [A] and mean [B] must decrease with the rise of mean [AB], the co-fluctuation of $[A]_t$ and $[B]_t$ could also be advantageous because it lowers the concentrations of the unbound monomers that may be toxic. Indeed, past studies found better expression co-fluctuations of genes encoding members of the same protein complex than random gene pairs [50, 51], suggesting that expression co-fluctuation of members of the same protein complex is selectively favored. Because our simulation considers protein concentrations instead of gene expressions, it directly applies to both haploid and diploid cells. The only difference is that protein concentrations such as $[A]_t$ and $[B]_t$ have a lower CV in diploid than haploid cells [6]. Further, as shown in S3 Fig, mouse cells analyzed here do show a higher correlation in the total expression level of two alleles between linked than unlinked genes.

To test if genes encoding components of the same protein complex tend to be linked, we used the mouse protein complex data from CORUM and downloaded the chromosomal positions of all mouse protein-coding genes from Ensembl [52]. Because genes may be linked due

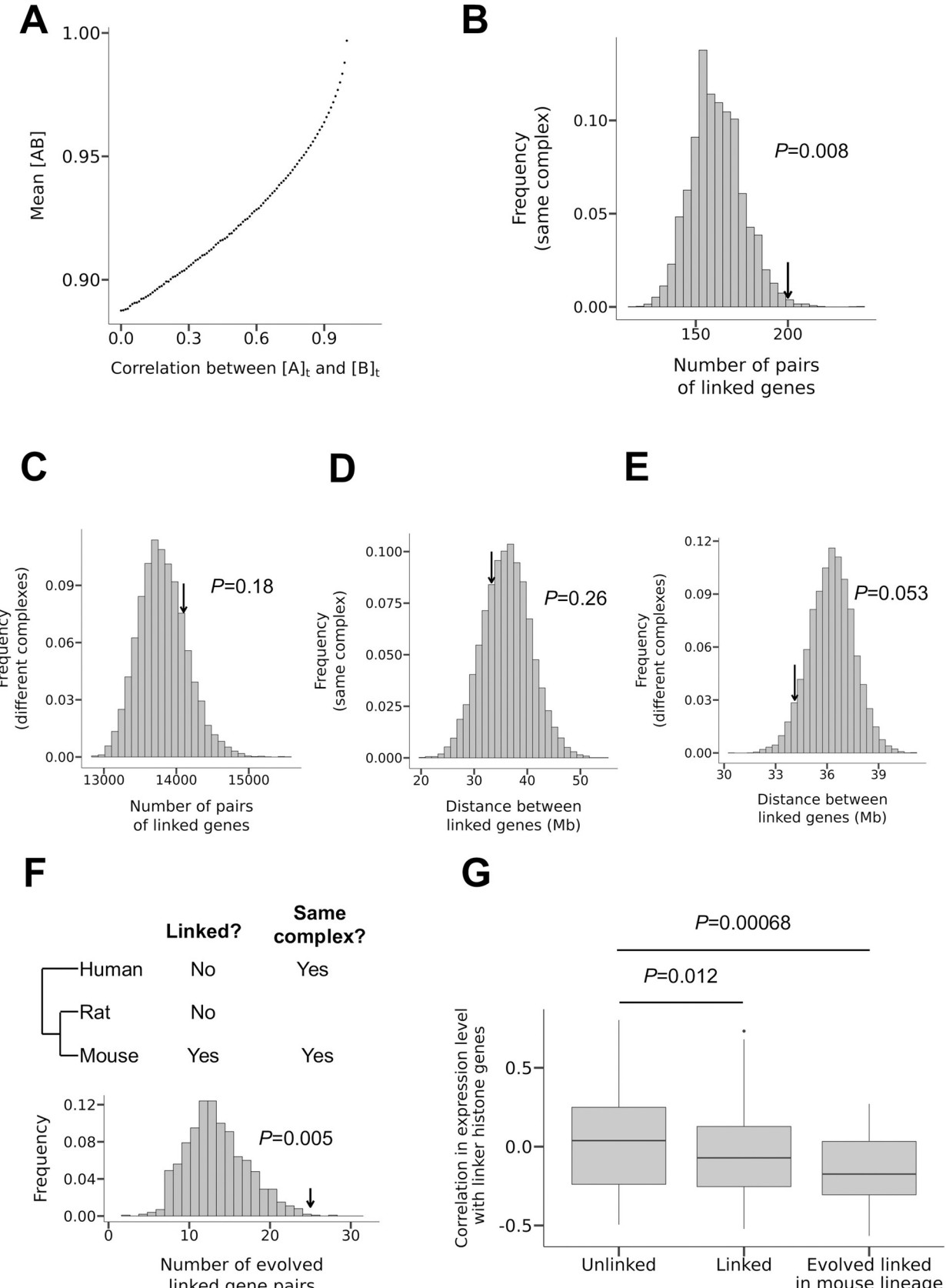

**Fig 4. Genes encoding components of the same protein complex tend to be chromosomally linked.** (A) Mean concentration of the protein complex AB ([AB]) in 10,000 cells increases with the co-fluctuation of the concentrations of its two components measured by the correlation of the total concentration of protein A ($[A]_t$) and that of B ($[B]_t$). (B-C) The frequency distribution of the number of pairs of linked genes encoding components of the same protein complex (B) and components of different protein complexes (C) in 10,000 randomly shuffled genomes. Arrows indicate the observed values. (D-E) The frequency distribution of the median distance between two linked genes that encode components of the same protein complex (D) and components of different protein complexes (E) in 10,000 randomly shuffled genomes. Arrows indicate the observed values. (F) Test of the hypothesis of protein complex-driven evolution of gene linkage, which asserts that the probability for an originally unlinked pair of genes to become linked is higher when they encode members of the same protein complex than when they do not. Of 875 pairs of genes that are unlinked in both human and rat and encode members of the same protein complex in both human and mouse, 25 become linked in mouse, as indicated by the arrow. The bars show the frequency distribution of the corresponding expected number under the null hypothesis. (G) Protein complex genes that are linked with at least one gene encoding a member of the same complex tend to be highly expressed in tissues with low abundances of linker histones. Y-axis shows the correlation in expression level between protein complex genes and the linker histone genes across tissues.

to their origins from tandem duplication [53], the data were pre-processed to produce a set of duplicate-free mouse protein-coding genes (see Materials and Methods). We then randomly shuffled the genomic positions of the retained genes encoding protein complex components among all possible positions of the duplicate-free mouse protein-coding genes. The observed number of linked pairs of genes encoding components of the same protein complex is significantly greater than the random expectation (Fig 4B). For comparison, we also computed the number of linked pairs of genes encoding components of different protein complexes. This number is not significantly greater than the random expectation (Fig 4C). Thus, the enrichment in gene linkage is specifically related to coding for subunits of the same protein complex. Interestingly, the observed median distance between the TSSs of two linked genes encoding protein complex subunits is not significantly different from the random expectation, regardless of whether subunits of the same (Fig 4D) or different (Fig 4E) protein complexes are considered.

The phenomenon that members of the same protein complex tend to be encoded by linked genes could have arisen for one or both of the following reasons. First, selection for co-fluctuation among proteins of the same complex has driven the evolution of gene linkage. Second, due to their co-fluctuation, products of linked genes may have been preferentially recruited to the same protein complex in evolution. Under the first hypothesis, originally unlinked genes encoding members of the same protein complex are more likely to become linked in evolution than originally unlinked genes that do not encode members of the same complex. To verify this prediction, we examined mouse genes using rat and human as outgroups (Fig 4F), because our $\delta_e$ estimates are from the mouse. We obtained pairs of genes encoding components of the same protein complex in both human and mouse. Hence, these pairs likely encode members of the same protein complex in the common ancestor of the three species. Among them, 875 pairs are unlinked in human and rat, suggesting that they were unlinked in the common ancestor of the three species. Of the 875 pairs, 25 pairs become linked in the mouse genome, significantly more than the random expectation under no requirement for gene pairs to encode members of the same complex ($P$ = 0.005; Fig 4F; see Materials and Methods). Therefore, the first hypothesis is supported. Under this hypothesis, the result in Fig 4D may be explained by the long-range linkage effect on expression co-fluctuation, such that once two genes encoding components of the same protein complex move to the same chromosome, selection is not strong enough to drive them closer to each other. To test the second hypothesis, we need gene pairs encoding proteins that belong to the same protein complex in mouse but not in human nor rat, which require such low false negative errors in protein complex identification that no current method can meet. Hence, we leave the test of the second hypothesis to future studies.

Note that the above tests have two caveats. First, it is possible that some tandem duplicates remain in our data, which will compromise the analysis in Fig 4B. However, the result in Fig

4F is robust to such potential errors because the gene pairs concerned are originally unlinked so could not have arisen by tandem duplication. Thus, our evidence for positive selection for gene linkage holds even when the data are contaminated by tandem duplicates. Second, we inferred ancestral gene linkage by the parsimony principle, which may occasionally be incorrect. But such errors add only random noise to our analysis, suggesting that the actual strength of evidence for our hypothesis is likely stronger than what is shown here.

As mentioned, our theoretical consideration suggests that, due to their intermediate diffusion coefficients, H1 histones may be responsible for the observed chromosome-wide expression co-fluctuation. Because the local H1 concentration fluctuates more when its cellular concentration is lower, we predict that the benefit and the selection coefficient for linkage of genes encoding members of the same protein complex are greater in tissues with lower H1 concentrations. Given that gene expression is costly, for a given gene, it is reasonable to assume that the relative importance of its function in a tissue increases with its expression level in the tissue [54, 55]. Hence, we predict that, the more negative the across-tissue expression correlation is between a protein complex member gene and H1 histones, the higher the likelihood that the gene is driven to be linked with other genes encoding members of the same protein complex. To verify the above prediction, we used a recently published RNA-seq dataset [56] to measure Pearson's correlation between the mRNA concentration of a gene that encodes a protein complex subunit and the mean mRNA concentration of all H1 histone genes across 13 mouse tissues. Indeed, the linked protein complex genes show more negative correlations than the unlinked protein complex genes ($P = 0.012$, one-tailed Mann-Whitney $U$ test; Fig 4G). The disparity is even more pronounced when we compare linked protein complex genes that become linked in the mouse lineage with unlinked protein complex genes ($P = 0.00068$, one-tailed Mann-Whitney $U$ test; Fig 4G). This is likely owing to the enrichment of genes that are linked due to the linkage effect in the group of evolved linked protein complex genes ($\frac{\text{Observed}-\text{null expectation}}{\text{null expectation}} = \frac{25-13}{13} = 92\%$) when compared with the group of linked protein complex genes ($\frac{\text{Observed}-\text{null expectation}}{\text{null expectation}} = \frac{200-161}{161} = 24\%$). The above three groups of genes (evolved linked protein complex genes, linked protein complex genes, and unlinked protein complex genes) were constructed using stratified sampling to ensure that their mean expression levels across tissues are not significantly different (see Materials and Methods). For comparison, we performed the same analysis but replaced H1 histones with TFIIB, a general transcription factor that is involved in the formation of the RNA polymerase II preinitiation complex and has a high diffusion rate [57]. The trends shown in Fig 4G no longer holds (unlinked vs. linked: $P = 0.11$, one-tailed Mann-Whitney $U$ test; unlinked vs. evolved linked: $P = 0.63$, one-tailed Mann-Whitney $U$ test). We also performed the same analysis but replaced H1 histones with core histone proteins, which are immobilized [45]. Again, the trends in Fig 4G disappeared (unlinked vs. linked: $P = 0.48$, one-tailed Mann-Whitney $U$ test; unlinked vs evolved linked: $P = 0.89$, one-tailed Mann-Whitney $U$ test). These results support our hypothesis about the role of H1 histones in the linkage effect of expression co-fluctuation.

## Discussion

Using allele-specific single-cell RNA-seq data, we discovered chromosome-wide expression co-fluctuation of linked genes in mammalian cells. We hypothesize and provide evidence that genes on the same chromosome tend to have close 3D proximity, which results in a shared chemical environment for transcription and leads to expression co-fluctuation. While the linkage effect on expression co-fluctuation is likely an intrinsic cellular property, when the expression co-fluctuation of certain genes improves fitness, natural selection may drive the relocation of these genes to the same chromosome. Indeed, we provide evidence suggesting

that the chromosomal linkage of genes encoding protein complex subunits is beneficial owing to the resultant expression co-fluctuation that minimizes the dosage imbalance among these subunits and has been selected for in genome evolution.

Although many statistical results in this study are highly significant, the effect sizes appear small in several analyses, most notably the $\delta_e$ and $\delta_a$ values for linked genes. The small effect sizes are generally due to the large noise in the data, less ideal types of data used, and mismatches between the data sets co-analyzed. For instance, $\delta_e$ between linked genes estimated here (Fig 2D) is much smaller than what was previously estimated for a pair of linked florescent protein genes [21], due in a large part to the inherently large error in quantifying mRNA concentrations by single-cell RNA-seq [58]. The small size of $\delta_a$ (Fig 3E) is likely caused at least in part by the low efficiency of ATAC-seq in detecting open chromatin (see Materials and Methods). The positive correlation between *trans-$r_a$* and *trans-$r_e$* (Fig 3G) is likely an underestimate due to the use of different cell types in RNA-seq and ATAC-seq. As shown in Fig 2E and 2F, the actual effect sizes would be much larger should better experimental methods and/or data become available. Hence, it is likely that many effects are underestimated in this study. Indeed, we estimated that the true effect sizes of $\delta_e$ and $\delta_a$ are at least an order of magnitude larger than observed (see Materials and Methods). In addition, the co-fluctuation effect detected by Raj et al. may be unusually large because in that study the chromosomal distance between the two genes was extremely small and the two genes used identical regulatory elements [21]. Regardless, we stress that whether an effect is large/important depends on whether it is detectable by natural selection, and our results in Fig 4 suggest that the effects appear visible to natural selection, as reflected in the preferential chromosomal linkage of genes encoding protein complex subunits. Note that natural selection can detect a selective differential as small as the inverse of the effective population size, which is about 70,000 in mouse [59].

Because the expression co-fluctuation of two genes can be achieved by the sharing of regulatory elements and/or linkage, it is important to understand the relative contributions of the two mechanisms. But this question is generally difficult to answer because the extent to which two genes share regulatory elements is usually unknown. However, a lower bound contribution of the linkage effect can be estimated by examining two (equally regulated) alleles of the same gene. In this extreme case, the contribution of the linkage effect on expression co-fluctuation is about one thirteenth the contribution of sharing regulatory elements (see Materials and Methods). Although linkage likely makes a smaller contribution than regulatory element sharing to the expression co-fluctuation of genes, linkage can increase the expression co-fluctuation to a level that regulatory element sharing alone cannot reach. This additional improvement can be important under certain circumstances, as shown recently for genes encoding enzymes of the yeast galactose use pathway [60].

Because we used RNA-seq to measure expression co-fluctuation, our results apply to the co-fluctuation of mRNA concentrations. In the case of protein complex components, it is presumably the co-fluctuation of protein concentrations rather than mRNA concentrations that is directly beneficial. Although the degree of covariation between mRNA and protein concentrations is under debate [61, 62], the two concentrations correlate well at the steady state [21]. One key factor in this correlation is the protein half-life, because, when the protein half-life is long, mRNA and protein concentrations may not correlate well due to the delay in the effect of a change in mRNA concentration on protein concentration [21]. It is interesting to note that in Raj et al.'s study [21], mRNA and protein concentrations still correlate reasonably well ($r = 0.43$) when the protein half-life is 25 hours, which is much longer than the reported mean protein half-life of 9 hours in mammalian cells [63]. Corroborating this finding is the recent report [64] that mRNA and protein concentrations correlate well across single cells in the

steady state (mean $r$ = 0.732). Note that, although the correlation between mRNA and protein concentrations measured at the same moment may not be high when the protein half-life is long, the current protein level can still correlate well with a past mRNA level [65]. Because our study focuses on cells at the steady state, co-fluctuation of mRNA concentrations is expected to lead to co-fluctuation of protein concentrations.

We attributed the preferential linkage of genes encoding protein complex subunits to the benefit of expression co-fluctuation, while a similar phenomenon of linkage was previously reported in yeast and attributed to the potential benefit of co-expression of protein complex subunits across environments [66], where co-expression refers to the correlation in mean expression level. In mammalian cells, our hypothesis is more plausible than the co-expression hypothesis for five reasons. First, across-environment (or among-tissue) variation in mean mRNA concentration does not translate well to the corresponding variation in mean protein concentration [62, 67], but mRNA concentration fluctuation explains protein concentration fluctuation quite well [21, 64]. Hence, gene linkage, which enhances mRNA concentration co-fluctuation and by extension protein concentration co-fluctuation, may not improve protein co-expression across environments. Second, co-expression of linked genes appears to occur at a much smaller genomic distance than the linkage effect on co-fluctuation detected here [68]. Thus, if selection on co-expression were the cause for the non-random distribution of protein complex genes, these genes should be closely linked. This, however, is not observed (Fig 4D). Hence, the previous finding that genes encoding members of (usually not the same) protein complexes tend to be clustered is best explained by the fact that certain chromosomal regions have inherently low expression noise and that these regions attract genes encoding protein complex members because stochastic expressions of these genes are especially harmful (i.e., the noise reduction hypothesis) [4, 69]. Third, the protein complex stoichiometry often differs among environments, which makes co-expression of complex components disfavored in the face of environmental changes [70, 71]. Nonetheless, under a given environment, protein concentration co-fluctuation remains beneficial because of the presence of an optimal stoichiometry at each steady state. Fourth, gene linkage is not necessary for the purpose of co-expression, because the genes involved can use similar *cis*-regulatory sequences to ensure co-expression even when they are unlinked. In fact, a large fraction of co-expression of linked genes is due to tandem duplicates [68], which have similar regulatory sequences by descent. However, even for genes with the same regulatory sequences, linkage improves expression co-fluctuation at the steady state. Finally, the co-expression hypothesis or noise reduction hypothesis cannot explain our observation of the relationship between the expression levels of H1 histones and those of linked genes encoding protein complex members across tissues (Fig 4G). Taken together, these considerations suggest that it is most likely the selection for expression co-fluctuation rather than co-expression across environments that has driven the evolution of linkage of genes encoding members of the same protein complex.

Several previous studies reported long-range coordination of gene expression [62, 72–79], but most of them was about co-expression above explained. One study used fluorescent in situ hybridization of intronic RNA to detect nascent transcripts in individual cells [72]. The authors reported independent transcriptions of most linked genes with the exception of two genes about 14 Mb apart that exhibit a negative correlation in transcription. Their observations are not contradictory to ours, because they measured the nearly instantaneous rate of transcription, whereas we measured the mRNA concentration that is the accumulated result of many transcriptional bursts. As explained, having a similar biochemical environment makes the activation/inactivation cycles of linked genes coordinated to some extent, even though the stochastic transcriptional bursts in the activation period may still look independent.

Our work suggests several future directions of research regarding expression co-fluctuation and its functional implications. First, it would be interesting to know if the linkage effect on expression co-fluctuation varies across chromosomes. Although we analyzed individual chromosomes (S5 Fig), addressing this question fully requires better single-cell expression data, because the current single-cell RNA-seq data are noisy. This also makes it difficult to detect any unusual chromosomal segment in its $\delta_e$ distribution. Second, our results suggest that 3D proximity is a major cause for the linkage effect on expression co-fluctuation. In particular, diffusion of proteins with intermediate diffusion coefficients such as H1 histones is likely one mechanistic basis of the effect. However, the diffusion behaviors of most proteins involved in transcription are largely unknown. A thorough research on the diffusion behaviors of proteins inside the nucleus will help us identify other proteins that are important in the linkage effect. As mentioned, our data do not allow a clear distinction between 3D looping and 3D diffusion in causing the linkage effect on tightly linked genes. To distinguish between these two mechanisms definitively, we would need allele-specific models of mouse chromosome conformation [80], which require more advanced algorithms and more sensitive allele-specific Hi-C methods. Third, our study highlights the importance of the impact of sub-nucleus spatial heterogeneity in gene expression. This can be studied more thoroughly via real-time imaging and spatial modeling of chemical reactions [43, 81]. The lack of knowledge about the details of transcription reactions prevents us from constructing an accurate quantitative model of gene expression, which can be achieved only by more accurate measurement and more advanced computational modeling. Fourth, we used protein complexes as an example to demonstrate how the linkage effect on expression co-fluctuation influences the evolution of gene order. Protein complex genes are by no means the only group of genes for which expression co-fluctuation can be advantageous. Previous work suggested that expression co-fluctuation of genes on the same signaling or metabolic pathway can be beneficial [82, 83], which was recently experimentally confirmed for the yeast galactose catabolism pathway [60]. But, to understand the broader evolutionary impact of the linkage effect, a general prediction of the fitness consequence of expression co-fluctuation is necessary. To achieve this goal, whole-cell modeling may be required [84]. Note that some other mechanisms such as cell cycle [85] can also lead to gene expression co-fluctuation, so should be considered in the study of the relationship between gene expression and fitness. Fifth, physical proximity might impact aspects of gene expression regulation other than co-fluctuation. For instance, previous research found that selective expression of genes that are clustered on the same chromosome (i.e., stochastic gene choice) is strongly dependent on intrachromosomal looping, which alters the pairwise physical distance between genes in the same gene array [86–88]. It will be interesting to explore whether the principle that governs the linkage effect studied here applies to stochastic gene choice. Sixth, because expression co-fluctuation could be beneficial or harmful, an alteration of expression co-fluctuation should be considered as a potential mechanism of disease caused by mutations that relocate genes in the genome. Seventh, our analysis focused on highly expressed genes more than lowly expressed ones due to the limited sensitivity of single-cell RNA-seq. Because lowly expressed genes are affected more than highly expressed genes by expression noise [89], expression co-fluctuation may be more important to lowly expressed genes than highly expressed ones. More sensitive and accurate single-cell expression profiling methods are needed to study the expression co-fluctuation of lowly expressed genes. Eighth, we focused on mouse fibroblast cells because of the limited availability of allele-specific single-cell RNA-seq data. To study how expression co-fluctuation impacts the evolution of gene order, it will be important to have data from multiple cell types and species. Last but not least, as we start designing and synthesizing genomes [90], it will be important to consider how gene order affects expression co-fluctuation and potentially fitness. It is possible that the fitness

effect associated with expression co-fluctuation is quite large when one compares an ideal gene order with a random one. It is our hope that our discovery will stimulate future researches in above areas.

## Materials and methods

### High-throughput sequencing data

The processed allele-specific single-cell RNA-seq data were downloaded from https://github.com/RickardSandberg/Reinius_et_al_Nature_Genetics_2016?files=1 (mouse.c57.counts.rds and mouse.cast.counts.rds). The Hi-C data [33] were downloaded from https://www.ncbi.nlm.nih.gov/geo/query/acc.cgi?acc=GSE72697, and we analyzed the 500kb-resolution Hi-C interaction matrix with high SNP density (iced-snpFiltered). The processed ATAC-seq data were provided by authors [39], and the data from 16 NPC populations were analyzed. All analyses were performed using custom programs in R or python.

### Protein complex data and pre-processing

The mouse protein complex data were downloaded from the CORUM database (http://mips.helmholtz-muenchen.de/corum/) [91]. The coordinates for all mouse protein-coding genes were downloaded from Ensembl BioMart (GRC38m.p5) [52]. To produce duplicate-free gene pairs, we also downloaded all paralogous gene pairs from Ensembl BioMart. Note that these gene pairs can be redundant, meaning that a gene may be paralogous with multiple other genes and appear in multiple gene pairs. We then iteratively removed duplicate genes based on the following rules. First, if one gene in a pair of duplicate genes has been removed, the other gene is retained. Second, if neither gene in a duplicate pair has been removed and neither encodes a protein complex component, one of them is randomly removed. Third, if neither gene in a duplicate pair has been removed and only one of them encodes a protein complex member, we remove the other gene. Fourth, if neither gene in a duplicate pair has been removed and both genes encode protein complex components, one of them is randomly removed. Applying the above rules resulted in a set of duplicate-free genes with as many of them encoding protein complex members as possible.

### Gibbs sampling for testing protein complex-driven evolution of gene order

We obtained all mouse genes that have one-to-one orthologs in both human and rat, and acquired from Ensembl their chromosomal locations in human, mouse, and rat. Gene pairs are formed if their products belong to the same protein complex in human as well as mouse, based on protein complex information in the CORUM database mentioned above. Among them, 875 gene pairs from 342 genes are unlinked in both human and rat, of which 25 pairs become linked in mouse. To test whether the number 25 is more than expected by chance, we compared these 342 genes with a random set of 342 genes that also form 875 unlinked gene pairs in human and rat. These unlinked pairs are highly unlikely to encode members of the same complex, so serve as a negative control. Because of the difficulty in randomly sampling 342 genes that form 875 unlinked gene pairs, we adopted Gibbs sampling [92], one kind of Markov-Chain Monte-Carlo sampling [93]. The procedure was as follows. Starting from the observed 342 genes, represented by the vector of (gene 1, gene 2, . . ., gene 342), we swapped gene 1 with a randomly picked gene from the mouse genome such that the 342 genes still satisfied all conditions of the original 342 genes described above. We then similarly swapped gene 2, gene 3, . . ., and finally gene 342, at which point a new gene set was produced. To allow the Markov chain to reach the stationary phase, we discarded the

first 1000 gene sets generated. Starting the 1001st gene set, we retained a set every 50 sets produced until 1000 sets were retained; this ensured relative independence among the 1000 retained sets. In each of these 1000 sets, we counted the number of gene pairs that are linked in mouse. The fraction of sets having the number equal to or greater than 25 was the probability reported in Fig 4F.

## Chromatin co-accessibility among cells vs. among cell populations

Let us consider the chromatin accessibilities of two genomic regions, A and B, in a population of N cells (N = 50,000 in the data analyzed) [39]. Let us denote the chromatin accessibilities for the two regions in cell i by random variables $A_i$ and $B_i$, respectively, where i = 1, 2, 3, ..., and N. We further denote the corresponding total accessibilities in the population as random variables AT and BT, respectively. We assume that $A_i$ follows the distribution X, while $B_i$ follows the distribution Y. We then have the following equations.

$$AT = \sum_{i=1}^{N} A_i \text{ and } BT = \sum_{i=1}^{N} B_i. \tag{1}$$

Pearson's correlation between AT and BT across cell populations all of size N is

$$Corr(AT, BT) = \frac{E(AT \cdot BT) - E(AT)E(BT)}{\sqrt{Var(AT)Var(BT)}} = \frac{E(\sum_{i=1}^{N} \sum_{j=1}^{N} A_i B_j) - N^2 E(X)E(Y)}{\sqrt{N^2 Var(X)Var(Y)}}$$

$$= \frac{\sum_{i=1}^{N} \sum_{j=1}^{N} E(A_i B_j) - N^2 E(X)E(Y)}{N\sqrt{Var(X)Var(Y)}}. \tag{2}$$

Because cells are independent from one another, when $i \neq j$,

$$E(A_i B_j) = E(A_i)E(B_j). \tag{3}$$

Thus,

$$\sum_{i=1}^{N} \sum_{j=1}^{N} E(A_i B_j) = \sum_{i=1}^{N} E(A_i B_i) + \sum_{i=1}^{N} \sum_{\substack{j=1 \\ j \neq i}}^{N} E(A_i)E(B_i)$$

$$= NE(XY) + (N^2 - N)E(X)E(Y). \tag{4}$$

Combining Eq (2) with Eq (4), we have

$$Corr(AT, BT) = \frac{NE(XY) - NE(X)E(Y)}{N\sqrt{Var(A) \cdot Var(B)}} = \frac{E(XY) - E(X)E(Y)}{\sqrt{Var(X) \cdot Var(Y)}} = Corr(X, Y). \tag{5}$$

Hence, if the number of cells per population is a constant and there is no measurement error, correlation of chromatin accessibilities of two loci among cells is expected to equal the correlation of total chromatin accessibilities per population of cells among cell populations.

To examine how violations of some of the above conditions affect the accuracy of Eq (5), we conducted computer simulations. We assume that the accessibility of a genomic region in a single cell is either 1 (accessible) or 0 (inaccessible). This assumption is supported by previous single-cell ATAC-seq data [40], where the number of reads mapped to each peak in a cell is nearly binary. Now let us consider two genomic regions whose chromatin states are denoted by A and B, respectively. The probabilities of the four possible states of this system are as

follows.

$$Pr(A = 0, B = 0) = p,$$
$$Pr(A = 0, B = 1) = q,$$
$$Pr(A = 1, B = 0) = r,$$
$$\text{and} \quad Pr(A = 1, B = 1) = s,$$

$(6)$

where $p + q + r + s = 1$. Hence, we have

$$E(A) = r + s,$$
$$E(B) = q + s,$$
$$E(AB) = s,$$
$$Var(A) = (r + s)(p + q),$$
$$Var(B) = (q + s)(p + r).$$

$(7)$

With Eq $(7)$, we can compute $Corr(A,B)$. In other words, for any given set of $p$, $q$, $r$, and $s$, we can compute the among-cell correlation in chromatin accessibility between the two regions.

We then generated 10,000 random sets of $p$, $q$, $r$, $s$ from a Dirichlet distribution. For each set of $p$, $q$, $r$, and $s$, we simulated the state of a cell by a random sampling from the four possible states. We did this for 16 cells as well as 16 cell populations each composed of 50,000 cells. We computed the total accessibility of each region in each cell population by summing up the corresponding accessibility of each cell. As expected, the among-cell correlation between the two regions in accessibility matches the true correlation (S6A Fig). The deviation from the true correlation is due to sampling error. Based on Eq $(5)$, the among-cell-population correlation between the two regions in total accessibility approximates the true correlation, which is indeed observed in our simulation (S6B Fig).

Nevertheless, accessibility of a region may be undetected due to low detection efficiencies of high-throughput methods, which makes the observed correlation between the accessibilities of two regions lower than the true correlation. To assess the impact of such low detection efficiencies on the correlation, we simulated a scenario with a 10% detection efficiency, which is common in high-throughput methods [58]. That is, for every accessible region, it is detected as accessible with a 10% chance and inaccessible with a 90% chance; every inaccessible region is detected as inaccessible with a 100% chance. Our simulation showed that the observed correlation between the accessibilities of two regions is weaker than the true correlation regardless of whether the data are from individual cells (S6C Fig) or cell populations (S6D Fig).

## True $\delta_a$ vs. observed $\delta_a$

The framework developed in the above section allows using computer simulation to acquire a lower-bound estimate of the true $\delta_a$. We simulated $\delta_a$ by considering two pairs of regions simultaneously. For each pair of regions, we first randomly sampled $p$, $q$, $r$, and $s$, followed by the computation of the true correlation using Eq $(7)$. The difference between the true correlations of the two pairs of regions is equivalent to the true $\delta_a$. Then, for each pair of regions, we can obtain the estimated $\delta_a$ from estimated correlations using simulation. In our allele-specific ATAC-seq data, only 55% of all reads are allele-specific. Given that in high-throughput sequencing data, the detection efficiency is 10 to 20% when all reads are considered [94], we choose 8.25% (= $0.15 \times 0.55$) as the detection efficiency in our simulation. We repeated this procedure 10,000 times and plotted the result in S7A Fig. We inferred the corresponding true $\delta_a$ from the observed median $\delta_a$ in the actual data using the regression in S7A Fig.

## True $\delta_e$ vs. observed $\delta_e$

To obtain a lower-bound estimate of the true $\delta_e$, we performed a simulation incorporating the known parameters of single-cell RNA-seq in our dataset. The simulation was performed as follows. First, we determined the mean expression levels for a pair of genes, $A$ and $B$, by sampling from the distribution of mean expression levels of genes analyzed, which was obtained based on the estimation that 1 RPKM corresponds to 1 transcript per cell in the original dataset [23]. Note that the mean expression level of each allele ($A_1$, $A_2$, $B_1$, and $B_2$) is one half the above sampled value. Second, we generated the expression levels across 60 cells for a pair of alleles ($A_1$ and $B_1$) from the joint multivariate normal distribution. The multivariate normal distribution can be uniquely determined once the correlation coefficient between the two alleles and their CV are chosen. We fixed the CV of the two alleles at 0.5, based on sm-FISH experiments in mammalian cells for genes whose expression levels are similar to the genes we analyzed [95]. Note that the CV used here is the mRNA CV, not the protein CV. The correlation between the two alleles was randomly sampled from the range (-1, 1). We name this correlation $r_1$. Third, for each allele in each cell, we used binomial sampling to determine the detected transcript level. In our data set, only 17% of the reads are allele-specific. Because the capturing efficiency is around 10–20% for full-length single cell RNA-seq data [94], we used 2.55% (= $0.15 \times 0.17$) as the sampling probability. Fourth, we computed the observed correlation between $A_1$ and $B_1$ across cells after binomial sampling. Fifth, we repeated the above steps 2 to 4. We named the newly sampled correlation $r_2$. The true $\delta_e$ would be $r_1 - r_2$, and the observed $\delta_e$ is the difference between the observed correlations. Sixth, we repeated 10,000 times steps 1 to 5, with all true $\delta_e$ and observed $\delta_e$ recorded (S7B Fig). We inferred the corresponding true $\delta_e$ from the observed median $\delta_e$ in the actual data using the regression in S7B Fig.

## Simulation of protein complex concentrations

Let the concentration of protein complex AB be [AB]. To study the average [AB] across cells in a population, we first simulated the concentrations of subunit A and subunit B in each cell. We assumed that the total concentrations of A and B, denoted by $[A]_t$ and $[B]_t$ respectively, are both normally distributed with mean = 1 and $CV = 0.2$. We used $CV = 0.2$ because this is the median expression noise measured by $CV$ for enzymes in yeast [6], the only eukaryote with genome-wide protein expression noise data [15]. Thus, the joint distribution of $[A]_t$ and $[B]_t$ is multivariate normal, which can be specified if the correlation ($r$) between $[A]_t$ and $[B]_t$ is known. With a given $r$, we simulated $[A]_t$ and $[B]_t$ for 10,000 cells by sampling from the joint distribution. We set the concentration to 0 if the simulated value is negative. We computed [AB] in each cell by solving the following set of equations.

$$[A]_t = [A] + [AB], \ [B]_t = [B] + [AB], \ \text{and} \ [AB] = K[A][B], \qquad (8)$$

where we used $K = 10^5$ based on the empirical values of association constants of stable protein complexes [49]. The mean complex concentration is the average [AB] among all cells. We also performed the simulation with other $K$ values ($10^{-2}$, $10^{-1}$, $10^0$, $10^1$, $10^2$, $10^3$, and $10^4$).

The above simulation can be extended for studying protein complexes with various stoichiometries. In general, for protein complex $A_M B_N$, we have

$$[A]_t = [A] + M[A_M B_N], \ [B]_t = [B] + N[A_M B_N], \ \text{and} \ [A_M B_N] = K[A]^M[B]^N. \qquad (9)$$

We considered $(M, N) = (1, 1)$, $(1, 2)$, $(1, 3)$, $(2, 2)$, $(2, 3)$, and $(3, 3)$, respectively.

In addition, we studied the consequence of having suboptimal mean $[A]_t$ or mean $[B]_t$. That is, we set the ratio of mean $[A]_t$ to mean $[B]_t$ at $M/N$, $2M/N$, or $0.5M/N$. We considered CV = 0.2 or 0.5.

## Relationship in expression level between protein complex genes and linker histone genes across tissues

This analysis used the RNA-seq data from 13 mouse tissues [56] as well as the protein complex data aforementioned. We divided all protein complex genes into three groups: unlinked genes, linked genes, and evolved linked genes. The first two groups are from duplicate-free protein complex gene pairs. A gene is assigned to the "linked" group if it is linked with at least one gene that encodes a member of the same protein complex. We found that the gene expression levels tend to be higher for the "linked" group than the "unlinked" group. To allow a fair comparison between these two groups, we computed the mean expression level of each gene across tissues and performed a stratified sampling as follows. We lumped all genes from the two groups and divided them into 20 bins based on their expression levels. For each bin, we counted the numbers of linked and unlinked genes respectively, and randomly down-sampled the larger group to the size of the smaller group. After the downsampling, the expression levels of the two groups of genes are comparable ($P$ = 0.9, two-tailed Mann-Whitney $U$ test). The third gene group contains genes that are linked in mouse but not in human nor in rat (i.e., "evolved linked"). We did not require them to be duplicate-free, but they were ancestrally unlinked so could not have resulted from tandem duplication. The expression levels of the third group of genes are not significantly different from those of the first two groups after the stratified sampling ($P$ = 0.68).

After obtaining the three groups of genes, we examined the among-tissue correlation between the expression level of each of these genes and the total expression level of all 11 H1 histone genes in mouse [96]. For control, we performed the same analysis but replaced H1 histones with TFIIB, a rapidly diffused transcription factor. In another control, we replaced H1 histones with immobilized core histones (H2A, H2B, H3, and H4). H2A, H2B, H3, and H4 genes are obtained from Mouse Genome Informatics (http://www.informatics.jax.org/) [97]:

http://www.informatics.jax.org/vocab/pirsf/PIRSF002048
http://www.informatics.jax.org/vocab/pirsf/PIRSF002050
http://www.informatics.jax.org/vocab/pirsf/PIRSF002051
http://www.informatics.jax.org/vocab/pirsf/PIRSF002052

## Contribution of the linkage effect relative to that of regulatory element sharing to expression co-fluctuation

The maximum effect of sharing regulatory elements on expression co-fluctuation, referred to as $\delta_{re}$, can be estimated by the median correlation coefficient in expression level between two alleles of the same gene minus the corresponding value for two alleles of different genes. We found that median $\delta_e$ is approximately one thirteenth of $\delta_{re}$. Thus, the linkage effect improves expression co-fluctuation brought by regulatory element sharing by at least one thirteenth.

## Data and software availability

Software used and the underlying numerical data for all figures can be downloaded from Github (https://github.com/mengysun/Linked_noise).

## Supporting information

**S1 Fig. Probability distributions of the correlation coefficient in expression level between alleles of two linked genes.** (A) The distribution of *cis*-correlations for alleles on C57BL/6J-derived chromosomes. (B) The distribution of *cis*-correlations for alleles on CAST/EiJ-derived chromosomes. (C) The distribution of *trans*-correlation for the C57BL/6J-derived allele at a gene and the CAST/EiJ-derived allele at another gene.
(PDF)

**S2 Fig. The linkage effect on expression co-fluctuation in clone 6 cells and non-clonal cells.** (A) Fraction of gene pairs with positive $\delta_e$ in clone 6. The red line represents the null expectation under no linkage effect. *P*-values from binomial tests on independent gene pairs are presented. (B) In clone 6, median $\delta_e$ in a bin decreases as the median genomic distance between linked genes in the bin rises. All bins have the same distance interval. TSS, transcription start site. The red line shows $\delta_e = 0$. The blue line shows the linear regression of binned data. Spearman's $\rho$ from unbinned data and associated *P*-value determined by a shuffling test are presented. (C) Fraction of gene pairs with positive $\delta_e$ in non-clonal mouse fibroblast cells. The red line represents the null expectation under no linkage effect. *P*-values from binomial tests on independent gene pairs are presented. (D) In non-clonal cells, median $\delta_e$ in a bin decreases as the median genomic distance between linked genes in the bin rises. All bins have the same distance interval. The red line shows $\delta_e = 0$. The blue line shows the linear regression of binned data. Spearman's $\rho$ from unbinned data and associated *P*-value determined by a shuffling test are presented.
(PDF)

**S3 Fig. The linkage effect on expression co-fluctuation in clone 7 cells analyzed using total reads of two alleles per locus.** (A) Median $\triangle_e$ in a bin decreases with the median genomic distance between linked genes in the bin. $\triangle_e$ for a linked gene pair is the correlation in RNA-seq read number between the two genes minus the median correlation for pairs of unlinked genes. All bins have the same distance interval. TSS, transcription start site. The red line shows $\triangle_e = 0$. The blue line shows the linear regression of binned data. Spearman's $\rho$ of unbinned data and associated *P*-value determined by a shuffling test are presented. (B) Median $\triangle'_e$ in a bin decreases with the corresponding median genomic distance between linked genes in the bin. $\triangle'_e$ for a linked gene pair is the correlation in expression level measured by RPKM (Reads Per Kilobase per Million mapped reads) between the two genes minus the corresponding median correlation for pairs of unlinked genes. The blue line shows the linear regression of binned data. Spearman's $\rho$ from unbinned data and associated *P*-value determined by a shuffling test are presented.
(PDF)

**S4 Fig. $\delta_e$ for pairs of neighboring genes with different orientations.** The lower and upper edges of a box represent the first (qu1) and third (qu3) quartiles, respectively, the horizontal line inside the box indicates the median (md), the whiskers extend to the most extreme values inside inner fences, md±1.5(qu3-qu1), and the dots represent values outside the inner fences (outliers). The nearest pairs were identified using the coordinates downloaded from Ensembl. After requiring a minimal read number of 10 for each allele, we separate neighboring gene pairs into three categories according to the orientations of their transcription directions. NS, $P > 0.05$, Wilcoxon rank-sum test.
(PDF)

**S5 Fig. $\delta_e$ decreases with distance between genes on each mouse chromosome.** Blue lines show linear regressions for binned data. All bins have the same distance intervals, while different chromosomes contain different numbers of bins depending on the chromosome length. Spearman's correlations from unbinned data and associated nominal $P$-values determined by shuffling tests are presented. Upon multiple testing correction, the correlations remain significant for chromosomes 1, 2, 5, 6, 11, and 12.
(PDF)

**S6 Fig. Simulation-based analysis of chromatin co-accessibility between two ATAC peaks quantified using single cells vs. using cell populations.** (A) The correlations quantified using single-cell-based measurements are close to their corresponding true correlations when the capturing efficiency is 100%. (B) The correlations quantified using cell-population-based measurements are close to the true correlations when the capturing efficiency is 100%. (C) The correlations quantified using single-cell-based measurements tend to be weaker than their corresponding true correlations when the capturing efficiency is 10%. (D) The correlations quantified using cell population-based measurements tend to be weaker than the true correlations when the capturing efficiency is 10%.
(PDF)

**S7 Fig. Simulations show that low capturing efficiencies lead to underestimation of $\delta_a$ and $\delta_e$.** (A) The magnitude of $\delta_a$ estimated from allele-specific ATAC-seq is much smaller than the true $\delta_a$. (B) The magnitude of $\delta_e$ estimated from allele-specific single-cell RNA-seq is much smaller than the true $\delta_e$.
(PDF)

## Acknowledgments

We thank members of the Zhang lab and three reviewers for valuable comments.

## Author Contributions

**Conceptualization:** Jianzhi Zhang.

**Data curation:** Mengyi Sun.

**Formal analysis:** Mengyi Sun.

**Funding acquisition:** Jianzhi Zhang.

**Investigation:** Mengyi Sun.

**Methodology:** Mengyi Sun.

**Project administration:** Jianzhi Zhang.

**Resources:** Jianzhi Zhang.

**Supervision:** Jianzhi Zhang.

**Writing – original draft:** Mengyi Sun, Jianzhi Zhang.

**Writing – review & editing:** Mengyi Sun, Jianzhi Zhang.

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
