## [Decision Letter · Decision Letter 0]

10 Jul 2019

Dear Dr Zhang:

Thank you very much for submitting your Research Article entitled 'Chromosome-wide co-fluctuation of stochastic gene expression in mammalian cells' to PLOS Genetics. Your manuscript was fully evaluated at the editorial level and by three independent peer reviewers. The reviewers appreciated the attention to an important topic but identified some aspects of the manuscript that should be improved.

We therefore ask you to modify the manuscript according to the review recommendations before we can consider your manuscript for acceptance. Your revisions should address the specific points made by each reviewer.

[LINK]

Yours sincerely,

Lu Bai

Guest Editor

PLOS Genetics

Bret Payseur

Section Editor: Evolution

PLOS Genetics

Reviewer's Responses to Questions

**Comments to the Authors:**

Reviewer #1: The study of Sun & Zhang reveals an interesting coupling between gene positioning along a chromosome and fluctuations, indicating that cis-correlations are higher than trans-correlations in the context of homologous chromosomes. The majority of tests are convincing, in particular the one showing that the effect becomes more prominent when they focus on highly expressed genes. The fact that the authors study the differential behavior of linked genes in comparison to their homologous counterparts (within a certain cell type) is clear in some but not in other contexts. Therefore, I suggest addressing the points below before the publication of the manuscript.

Major remarks

1) The comparison of expression fluctuations with chromosomal looping and chromatin accessibility clearly follows physical models of gene expression. However, the hypothesized link between expression fluctuations and protein complexes “ the expression co-fluctuation of genes encoding components of the same protein complex is likely advantageous.” should be examined in more detail. The authors implicitly assume the formation of protein complexes is enhanced when the subunits expressed in a coordinated fashion, irrespective of the basis of comparison. However, it is possible that there is no difference in the correlation with homologous and heterologous interchromosomal fluctuations. This is important since each subunit of the dimer is expressed from a maternal and paternal chromosome even if they are positioned on homologous and heterologous chromosomes. Therefore, either the simulations should include different scenarios of chromosomal fluctuations or the above conclusion should be explained statistically more rigorously.

Minor remarks:

2) A. The authors compare stochastic intra-chromosomal and inter-chromosomal correlations. This is important because most studies (as mentioned in the discussion) do not distinguish intra-chromosomal effect. This is mentioned at several places, notably in the abstract, in the last paragraph of the introduction , first section of the results,…. However, the authors should not assume that the general audience understands the term “linked” and should explain that in more general terms in the author summary. For example, the authors could explain in the author summary that each individual receives a maternal and paternal copy of the same chromosome.

B. Figure 1 or Figure 2A should include the full depiction of pair(s) of homologous chromosomes so that it becomes clear what the authors mean.

3) The interchromosomal fluctuations may have a substantially stronger effect on gene choice than on protein complexes. Therefore, I suggest the authors discussing stochastic gene choice in the context of their results. The Protocadhering gene array is prototypical example where stochastic gene expression is very strongly depends on intrachromosomal looping controlled by the binding of the looping factor CTCF in neuronal progenitors (PMID: 15640798, 30257211). Therefore, the intrachromosomal fluctuations and conformational changes may have a strong impact on the choice and number of expressed protein isoforms of this membrane protein, which can underscore the biological relevance of the fluctuations studied by the authors.

Reviewer #2: RE: PGENETICS-D-19-00916

In this study, the authors investigated co-fluctuation of chromosomally-linked genes. This phenomenon was first discovered experimentally by a previous study using fluorescent reporters (Raj et al, 2006). Here the authors analyzed the existing single-cell RNA seq data and showed that this could be a genome-wide phenomenon. To this end, they showed statistically the expression correlations between linked alleles (on the same chromosome) of two genes are greater than those between unlinked alleles. With the support of additional analysis, they further demonstrated that the 3D proximity and chromatin co-accessibility could be the underlying mechanism of this effect. Finally, they proposed that, through natural selection, genes encoding components of the same complexes might be organized onto the same chromosomes to correlate their expression.

I find this paper well-written and scientifically interesting. Gene expression coordination in single cells has been an important topic of research. The authors performed a careful analysis of existing data and raised an intriguing model. I would recommend the acceptance of this manuscript for publication shall the concerns below be addressed.

1. I would think that the most direct way to coordinate gene expression of functionally-related genes in individual cells is through sharing the same trans or cis-regulatory factors. Is there a way to quantitively compare the contribution of this mechanism with the contribution of chromosomal linkage? I suspect that the former one might play a major role. The paper is interesting either way, but a careful discussion and comparison will be helpful.

2. Related to point 1, for natural selection, would sharing regulatory elements provide more plasticity than reorganizing the genome? Again, a careful discussion would be needed. Genome re-organization does not seem like the more intuitive way to ensure gene expression coordination.

3. In addition to genes encoding components of the same complexes, I wonder whether the same organization rule can be applied to the genes encoding components of the same signaling or metabolic pathways? If not, why?

Reviewer #3: The manuscript from Mengyi Sun and Jianzhi Zhang lays out the novel and provocative idea that coordinated fluctuations in gene expression stem from fluctuations in local histone H1 concentrations.

In support of this idea, they show that expression of "linked" (immediate nearest neighbors along the same chromosome) genes is more correlated than expression of the same pair of genes across homologous chromosomes. Suprisingly, this is true even when the two genes are seperated by over 100Mb, suggesting a 3D, not 1D, cause. Consistent with this, but unsuprisingly, alleles on the same chrommosome are more frequently in physical contact with each other than alleles on homologous chromosomes.

As a mechanism they propose local diffusion of H1, and in support of this mechanism they find that protein complex members whose expression is negativly correlated with H1 mRNA levels are more likely to be linked.

The ideas are provocative and, while, as the authors point out, the effect sizes are weak, some of this may be due to imperfect matches between experimental data and experimental error. In addition, evolution can act on weak effect sizes.

However, their model makes a major prediction that I'm suprised they didn't test, and I would like to see tested before the paper is published. The authors propose that 1D distance doesn't matter; only 3D distance. This predicts that the correlation shown in Fig1D should be even stronger if 3D distance is plotted on the X axis. Furthremore, being nearest neighbors (linked) shouldn't be the most predictive feature. HiC contact frequency should be.

If you recalculate something like Figure 1B but bin into multiple groups, binning genes by contact frequency and/or distance in 3D space, instead of only into linked vs unlinked, does the effect still hold?

In other words, genes that are unlinked, but still close in 3D space, should exhibit equally strong co-expression.

Minor Point:

"While each of the four correlations could be positive or negative, in the large data analyzed below, they are mostly positive and show approximately normal distributions across gene pairs examined."

I'd like to see the histograms of these four correlations (re(A1,B1), re(A2,B2), etc)

"All statistical analyses were performed using custom R and python scripts that are available upon request."

It's 2019; this should no longer be acceptable. Please put all code and intermediate (processed) data on github or some other repository.

The PLoS Genetics data availibility guidelines state that, if the data are not made publically availible, "The reasons for restrictions on public data deposition must also be specified. "

**Have all data underlying the figures and results presented in the manuscript been provided?**

Reviewer #1: Yes

Reviewer #2: None

Reviewer #3: No: "All statistical analyses were performed using custom R and python scripts that are available upon request."

It's 2019; this should no longer be acceptable. Please put all code and intermediate (processed) data on github or some other repository.

The PLoS Genetics data availibility guidelines state that, if the data are not made publically availible, "The reasons for restrictions on public data deposition must also be specified. "

PLOS authors have the option to publish the peer review history of their article (what does this mean?). If published, this will include your full peer review and any attached files.

Reviewer #1: No

Reviewer #2: No

Reviewer #3: Yes: Lucas Carey

---

## [Decision Letter · Decision Letter 1]

28 Aug 2019

[EXSCINDED]

Dear Dr Zhang,

We are pleased to inform you that your manuscript entitled "Chromosome-wide co-fluctuation of stochastic gene expression in mammalian cells" has been editorially accepted for publication in PLOS Genetics. Congratulations!

There are some minor comments from Reviewer #1 about wording and references, please take that into consideration for the finalized version. Before your submission can be formally accepted and sent to production you will need to complete our formatting changes, which you will receive in a follow up email. Please be aware that it may take several days for you to receive this email; during this time no action is required by you. Please note: the accept date on your published article will reflect the date of this provisional accept, but your manuscript will not be scheduled for publication until the required changes have been made.

Yours sincerely,

Lu Bai

Guest Editor

PLOS Genetics

Bret Payseur

Section Editor: Evolution

PLOS Genetics

Comments from the reviewers (if applicable):

Reviewer's Responses to Questions

**Comments to the Authors:**

Reviewer #1: The authors addressed most of the points raised. In particular, they rewrote the author summary and extended Figure 2 to facilitate readability. They also inserted a brief discussion that the baseline protein noise level may vary with the genomic framework (e..g. haploid and diploid) but this does not change the conclusions. They cite a relevant paper in this context.

They also discuss the looping in the context of gene choice “For instance, previous research found that selective expression of genes that are clustered on the same chromosome (i.e., gene choice) is strongly dependent on intrachromosomal looping, which alters the pairwise physical distance between genes in the same gene array.“ It would be more precise to state “stochastic gene choice” than merely “gene choice”. Importantly, the authors omitted to cite the relevant papers to specify the previous research:

Please add these references so that the meaning of these sentences becomes clear.

“Monoallelic yet combinatorial expression of variable exons of the protocadherin-alpha gene cluster in single neurons.”

“Stochastic Gene Choice during Cellular Differentiation.”

“CTCF/cohesin-mediated DNA looping is required for protocadherin α promoter choice.”

These three papers provide preliminary evidence that looping drives stochastic gene choice that the expression frequency correlates with the binding of the looping factor, CTCF, to the respective genes.

Reviewer #2: The revisions have solved all my concerns. I recommend acceptance of this paper for publication.

**Have all data underlying the figures and results presented in the manuscript been provided?**

Reviewer #1: Yes

Reviewer #2: Yes

PLOS authors have the option to publish the peer review history of their article (what does this mean?). If published, this will include your full peer review and any attached files.

Reviewer #1: No

Reviewer #2: No

**Data Deposition**

http://datadryad.org/submit?journalID=pgenetics&manu=PGENETICS-D-19-00916R1

**Press Queries**

---

## [Editor Report · Acceptance letter]

10 Sep 2019

PGENETICS-D-19-00916R1 

Chromosome-wide co-fluctuation of stochastic gene expression in mammalian cells 

Dear Dr Zhang, 

We are pleased to inform you that your manuscript entitled "Chromosome-wide co-fluctuation of stochastic gene expression in mammalian cells" has been formally accepted for publication in PLOS Genetics! Your manuscript is now with our production department and you will be notified of the publication date in due course.

With kind regards,

Kaitlin Butler

PLOS Genetics

On behalf of:
